# Diffusion-based Spatio-temporal Interpolation with Dynamic Sensor Sets

## Abstract

We tackle spatio-temporal interpolation for virtual sensors in sparse, partially observed, and dynamically changing networks. We introduce DynaSTI, a diffusion-based generative framework that is fully inductive to unseen locations, trains directly on incomplete observations, and remains effective without retraining when sensor networks change with time. Our contributions are threefold: (i) a unified conditioning strategy that yields calibrated predictive distributions and robust performance under severe input-sensor dropout; (ii) a Fourier-domain compression variant, FDynaSTI, that accelerates sampling performance, and (iii) state-of-the-art performance on multiple real-world datasets, improving both RMSE and CRPS relative to strong baselines. Together, these results establish diffusion-based, frequency-aware probabilistic interpolation as a scalable solution for real-world, dynamic sensor networks.

## 1 Introduction

Accurate modeling and interpolation of spatio-temporal signals underpin a wide range of practical applications, including environmental monitoring, traffic analysis, and urban planning. A particularly challenging yet important task within this domain is predicting multivariate time-series data at locations devoid of sensors or historical observations, commonly termed as virtual sensors. This process, known as spatio-temporal interpolation, is essential for enabling informed decision making in scenarios where sensor networks are sparse, incomplete, or subject to dynamic changes. Our primary objective is to develop a robust method that accurately predicts data for these unseen locations without requiring prior knowledge of their coordinates during training. Our model is specifically engineered to handle sensor networks that change over time, while assuming only that the underlying observation distribution remains stationary.

Classical approaches such as kriging/Gaussian processes (Matheron, 1963; Cressie, 1993; Rasmussen & Williams, 2006; Hamelijnck et al., 2021), ARMA/VAR models (Box & Jenkins, 1990), EM-based matrix/tensor completion (Dempster et al., 1977), and spatial statistics provide principled uncertainty but often rely on strong stationarity assumptions, hand-crafted kernels, dense coverage, and fixed topologies; they also struggle to scale to large, high-dimensional networks. More recent deep methods—graph neural networks (Cini et al., 2022; Tharzeen et al., 2023; Yang et al., 2025; Kuppannagari et al., 2021), attention-based sequence models (Marisca et al., 2022; Nie et al., 2024), neural processes (Hu et al., 2023), and diffusion-based imputers (Liu et al., 2023; Hu et al., 2024; Tashiro et al., 2021; Islam et al., 2025)–have improved expressivity and accuracy on partially observed data, yet commonly presume that the set of observed and target sensors and their graph/topology are fixed at training and test time. Moreover, uncertainty quantification for unseen locations is frequently ad hoc (e.g., MC-dropout (Gal & Ghahramani, 2016)) or absent, and many models require costly retraining or architectural change when the sensors drop out. In summary, no existing approach simultaneously supports dynamic sensor configurations, long sequences, multivariate data, and uncertainty prediction for virtual sensors.

To address these challenges, we propose a novel diffusion-based generative framework for spatio-temporal interpolation that leverages spatial, temporal, and feature encoders. Our model, *DynaSTI*, short for "**Dyna**mic **S**patio-**T**emporal **I**nterpolation via Diffusion", is specifically engineered to predict multivariate time-series data at virtual sensor locations, excelling in scenarios where input sensor data may be incomplete or missing at training and testing time. Moreover, Our model eliminates

Table 1: Comparison of spatio-temporal interpolation methods by key strengths.

| Strength | GRIN | ST-GAIN | SPIN | PriSTI | IGNNK | DeepKriging | ST-VGP | KITS | GSLI | BayesNF | MMGN | USTD | ImputeFormer | DynaSTI |
|---|---|---|---|---|---|---|---|---|---|---|---|---|---|---|
| Dynamic Topology | | | | | ✓ | | ✓ | ✓ | ✓ | ✓ | ✓ | ✓ | | ✓ |
| Inductive | | | | | ✓ | ✓ | ✓ | ✓ | ✓ | ✓ | ✓ | ✓ | | ✓ |
| Generative | | ✓ | | ✓ | | | ✓ | | | ✓ | ✓ | ✓ | | ✓ |
| Multivariate | ✓ | ✓ | ✓ | ✓ | ✓ | ✓ | ✓ | ✓ | ✓ | | ✓ | ✓ | ✓ | ✓ |
| Incomplete inputs | ✓ | ✓ | ✓ | ✓ | ✓ | | | ✓ | ✓ | ✓ | ✓ | ✓ | ✓ | ✓ |

the need for retraining when sensor configurations change, is fully inductive to unseen locations, uses a unified conditioning scheme to ingest whatever sensors are available, naturally represents uncertainty, and accelerates long-horizon inference through a Fourier-domain compression–together yielding a scalable, and cost-effective solution.

Table 1 summarizes how various models for spatio-temporal interpolation address four critical strengths relevant to virtual sensor prediction – (1) **Dynamic Topology**, which is adaptability to sensor networks whose configuration changes dynamically (e.g., sensors being added or removed), (2) **Inductive** is the ability to predict at locations unseen during training, (3) **Generative** is to providing probabilistic outputs rather than solely deterministic predictions, (4) **Multivariate** is the ability to predict for multivariate data instead of just univariate, and (5) **Incomplete inputs** is the ability to handle incomplete data with arbitrary missing feature values as input at training and testing time. While USTD (Hu et al., 2024) meets all criteria in Table 1, the public implementation restricts sequences to 12 or 24 steps, so we were unable to run it on our datasets, which have longer sequences.

Our key contributions are:

- **Inductive diffusion for virtual sensors:** We propose a fully inductive, diffusion-based framework that trains directly on incomplete data and generalizes to unseen target locations without retraining.

- **Unified conditioning with locations and probabilistic prediction:** We introduce a conditioning strategy that integrates irregular spatio-temporal context (locations) into the denoising process, delivering robust performance under severe sensor dropout. The approach produces probabilistic predictions yielding uncertainty quantification.

- **Fourier-domain compression for long sequences:** We develop a frequency-aware representation that expresses each series as a trend (intercept + slope) and seasonality to accelerate the inference time.

- **Strong empirical performance and robustness:** Across diverse, real-world datasets, DynaSTI achieves state-of-the-art accuracy relative to strong baselines (e.g., lower RMSE and improved CRPS) and shows graceful degradation with sensor dropout.

## 2 RELATED WORK

**Traditional statistical approach:** Classical methods—ARMA (Box & Jenkins, 1990), EM (Dempster et al., 1977), and KNN (Fix & Hodges, 1951)—leverage temporal smoothness or spatial similarity but miss complex dependencies. Kriging (Matheron, 1963; Cressie, 1993) and Gaussian Processes (GPs) (Rasmussen & Williams, 2006; Cressie & Wikle, 2011) provide principled uncertainty via covariance kernels, yet suffer from cubic scaling and sensitivity to stationarity/kernel choice.

**Deterministic deep models:** Graph-based models (GRIN (Cini et al., 2022), GLSTM (Tharzeen et al., 2023), STGNN-DAE (Kuppannagari et al., 2021)) capture spatial–temporal structure but typically assume fixed topologies and can accumulate autoregressive errors in sparse regimes. GSLI (Yang et al., 2025) learns multi-scale graphs to handle node/feature heterogeneity but adds compute overhead. Attention models like SPIN (Marisca et al., 2022) enable virtual prediction with local representations but require known locations during training. Other deterministic methods–tensor completion (Ben Said & Erradi, 2022; Zhang & Wei, 2024), MLP-RAIN (Saubhagya et al., 2024), and image inpainting Yun et al. (2023)–work well on grids yet adapt poorly to irregular/dynamic graphs and usually lack calibrated uncertainty. Inductive kriging with GNNs (IGNNK (Wu et al., 2021), DeepKriging (Nag et al., 2023), INCREASE (Zheng et al., 2023), KITS (Li et al., 2023)) generalizes to unseen nodes but often simplifies temporal dynamics, may depend on side information,

or rely on pseudo-labels. INR-style continuous fields (MMGN) (Luo et al., 2024) learns coordinate-to-value mappings but is deterministic and lacks explicit graph inductive bias. STFNN (Feng et al., 2024) models air-quality data as a continuous spatio-temporal field, learning its gradient with implicit neural representations and refining predictions through Pyramidal Inference to capture both low- and high-frequency patterns. Casper (Jing et al., 2024) introduces a causality-aware STGNN that uses frontdoor adjustment, a Prompt-Based Decoder, and Spatiotemporal Causal Attention to mitigate spurious correlations and learn sparse causal relations.

**Probabilistic methods:** Probabilistic approaches quantify uncertainty: STGNP (Hu et al., 2023) (neural processes on graphs) relies on predefined graphs/covariates. Bayesian Neural Fields (Saad et al., 2024) scale via hierarchical inference but approximate posteriors. ST-VGP (Hamelijnck et al., 2021) uses variational/state-space structure for linear-time scaling with assumptions on kernels/likelihoods. ST-GAIN (Zhang et al., 2017) is a GAN-based imputation model, which suffers from the training instability of GANs. Diffusion models handle nonstationarity generatively: CSDI (Tashiro et al., 2021) and SADI (Islam et al., 2025) impute time series without spatial context; PriSTI (Liu et al., 2023) adds spatio-temporal conditioning but remains non-inductive; USTD (Hu et al., 2024) unifies forecasting/kriging with a shared encoder and gated-attention decoders, supports inductive kriging but is evaluated on fixed training graphs and short sequences; VDM (Li et al., 2026) combines VAE pre-imputation, multi-scale trends, and temporal Mamba (Gu & Dao, 2024) with dynamic/static graph encoders, assuming a fixed distance graph and risking VAE bias.

## 3 PRELIMINARIES

In this section, we first outline our problem setup and then provide an overview of the diffusion model concepts that are relevant to our approach.

### 3.1 PROBLEM SETUP

We tackle spatio-temporal interpolation, predicting target time-series at arbitrary locations from multivariate sensors with missing and time-varying observations. This reflects real deployments (digital agriculture, atmospheric sensing) where sparse networks face outages and changing topologies. Our goal is a single model that generalizes across variables, time intervals, and virtual locations using the available sensor data.

More formally, a spatio-temporal dataset consists of sensor observations indexed by location and time. We let $\mathcal{S}$ denote the set of possible spatial coordinates, which will typically be 2D or 3D geographic locations, and $\mathcal{S}'$ denote the finite set of locations involved in the data under consideration. We consider a discrete time model where $\mathcal{T} = \{t_1, t_2, \ldots, t_T\}$ is the set of regularly sampled time steps spanning the temporal extent of the data. We consider multivariate sensors with $C$ channels that produce data of the form $(s, t, \mathbf{x}, \mathbf{m})$, indicating the sensor location $s \in \mathcal{S}'$, the measurement time $t \in \mathcal{T}$, sensor values $\mathbf{x} \in \mathbb{R}^C$, and a channel mask $\mathbf{m} \in \{0, 1\}^C$, indicating which channels are missing (0 indicates missing). Importantly, we make no assumptions about how many missing values are in the data at either training or testing time. The observed dataset is defined as:

$$\mathcal{D} = \left\{ (s, t, \mathbf{x}, \mathbf{m}) \;\middle|\; s \in \mathcal{S}', t \in \mathcal{T}, \mathbf{x} \in \mathbb{R}^C, \mathbf{m} \in \{0, 1\}^C, \text{ where } m_c = 1 \text{ if } x_c \text{ is observed, else } 0 \right\}$$

The goal of spatio-temporal interpolation is to estimate one or more unobserved sensor channels at an arbitrary location $s^*$ and all time steps $\mathcal{T}$ using a dataset of observed data $\mathcal{D}$. This process handles situations where some channel values are observed at $s^*$, with a mask $M^* \in \{0, 1\}^{T \times C}$, which indicates which features/channels in $X^* \in \mathbb{R}^{T \times C}$ are missing for any time and are to be estimated as the data at the target location can be partially observed. For virtual locations, the mask $M^*$ is all zeros and the goal is to predict the entire time-series data at the location $s^*$.

### 3.2 DIFFUSION MODELS

Diffusion models provide a generative framework for sampling from complex data distributions. They define a *forward diffusion* process that incrementally adds noise to data, and a *reverse denoising* process that learns to remove this added noise. Concretely, let $X_0 \sim q(X_0)$ be a sample from the true data distribution. The forward diffusion process is a fixed Markov chain:

$$q\big(X_k \mid X_{k-1}\big) = \mathcal{N}\Big(X_k; \; \sqrt{1 - \beta_k}\, X_{k-1}, \; \beta_k \, \mathbf{I}\Big), \quad k = 1, \ldots, K,$$

where $\beta_k$ is a variance schedule and $\mathbf{I}$ is the identity matrix. After $K$ steps, $X_K$ is nearly isotropic Gaussian noise.

A *reverse diffusion* model $p_\theta$ parameterized by $\theta$ is then trained to reconstruct $X_0$ from $X_K$ via:

$$p_\theta\big(X_{k-1} \mid X_k\big) = \mathcal{N}\big(X_{k-1}; \boldsymbol{\mu}_\theta(X_k, k), \boldsymbol{\Sigma}_\theta(X_k, k)\big).$$

Sampling from the trained model amounts to starting from Gaussian noise $X_K \sim \mathcal{N}(0, \mathbf{I})$ and iteratively applying the learned reverse steps:

$$X_{k-1} \sim p_\theta\big(X_{k-1} \mid X_k\big), \quad k = K, K-1, \ldots, 1.$$

A popular way to train the reverse diffusion model is via *noise prediction*. Let $\bar{\alpha}_k = \prod_{\ell=1}^{k}(1 - \beta_\ell)$. We construct $X_k$ by mixing the clean sample $X_0$ with Gaussian noise $\varepsilon$:

$$X_k = \sqrt{\bar{\alpha}_k}\, X_0 \;+\; \sqrt{1 - \bar{\alpha}_k}\, \varepsilon, \quad \varepsilon \sim \mathcal{N}(0, \mathbf{I}).$$

The model $p_\theta$ (often denoted $\varepsilon_\theta$) directly predicts $\varepsilon$. This yields a simple mean-squared error objective according to Ho et al. (2020):

$$\mathcal{L}(\theta) = \mathbb{E}_{x_0 \sim q,\, \varepsilon \sim \mathcal{N}(0, \mathbf{I}),\, k \sim \mathrm{Uniform}\{1, \ldots, K\}} \Big[ \left\| \varepsilon - \varepsilon_\theta(X_k, k) \right\|^2 \Big].$$

Minimizing this loss encourages $\varepsilon_\theta$ to correctly denoise $X_k$ at each step $k$. By integrating conditional mechanisms into $\varepsilon_\theta$, these models can be adapted for *conditioned generation*, making them suitable for tasks such as spatio-temporal interpolation. In such a setup, the model conditions on partial observations (spatial locations, diffusion steps, known features) and generates the missing values accordingly.

# 4 DYNAMIC SPATIO-TEMPORAL DATA INTERPOLATION

We propose DynaSTI, a DDPM-based (Ho et al., 2020) framework for multivariate spatio-temporal interpolation that predicts time series at virtual sensor locations without prior coordinate knowledge. It conditions noise prediction on observed measurements, sensor coordinates, and the diffusion step–naturally handling missing data and dynamic topologies. The denoiser comprises three modules: Spatial, Temporal, and Feature Encoders that capture spatial correlations, long-range temporal dependencies, and cross-channel interactions, respectively (Fig. 1).

## 4.1 MODEL OVERVIEW

Our model operates within the DDPM framework, where a forward diffusion process incrementally adds Gaussian noise to the data, and a reverse denoising process learns to reconstruct the original data distribution. For spatio-temporal interpolation, we aim to estimate multivariate time-series data $X^* \in \mathbb{R}^{T \times C}$ at a target location $s^* \in \mathbb{R}^d$ and time steps $\mathcal{T}$, conditioned on observed measurements $\mathcal{D}$ (See Section 3.1). The denoising model $\varepsilon_\theta$ estimates the noise $\varepsilon$ added to the target data, conditioned on the noisy target data $X_k^* \in \mathbb{R}^{T \times C}$, the target location $s^*$, the observed data along with their sensor location $\mathcal{D}$ and the diffusion step $k$. Two binary masks manage missing data: one for features at observed locations and another for target locations, with $1$ denoting observed data and $0$ denoting missing data. The model is trained to minimize the noise prediction loss:

$$\mathcal{L}(\theta) = \mathbb{E}_{X_0^* \sim q, \varepsilon \sim \mathcal{N}(0, \mathbf{I}), k \sim \mathrm{Uniform}(1, \ldots, K)} \Big[ \|\varepsilon - \varepsilon_\theta(X_k^*, k, s^*, \mathcal{D})\|^2 \Big] \tag{1}$$

The denoising model integrates three specialized encoders to model the complex dependencies in spatio-temporal data. These encoders are applied sequentially, with multiple layers of each to refine the representations, and the diffusion step $k$ is embedded as a conditioning signal to guide the denoising process. The following subsections detail each component, the Fourier compression upgrade, and the training/inference pipeline.

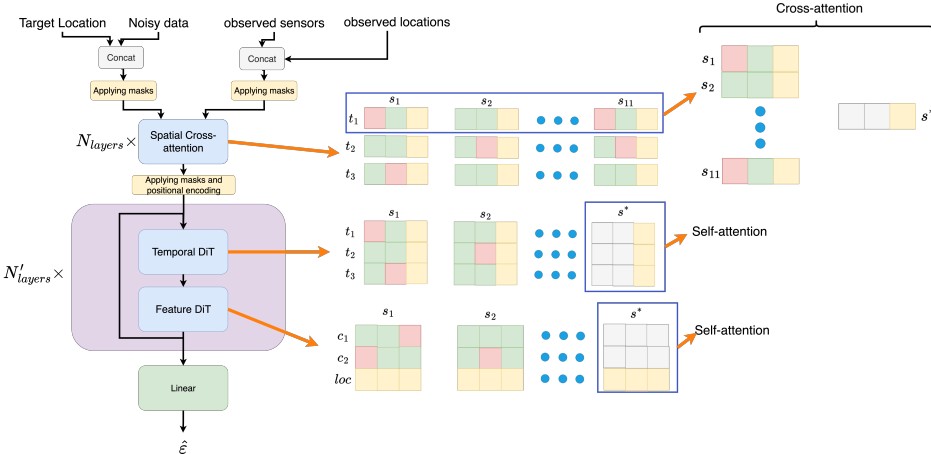

Figure 1: Overview of the model architecture when predicting the time-series data at one target location $s^*$ (where there is no sensor) given 11 observed locations, each with two features. The green cells denote observed values at each sensor, red cells indicate missing entries, yellow cells store location metadata, and the grey cell represents a noisy placeholder for the target location's missing time series. The model applies spatial cross-attention on the observed sensors, followed by dedicated temporal and feature "DiT" modules, ultimately producing a prediction for the target location from the blended information of other neighbors.

## 4.2 SPATIAL ENCODER

The Spatial Encoder models the spatial relationships between the target location $s^*$ and the observed sensor locations, while incorporating the diffusion step $k$ to condition the denoising process. To handle incomplete data, we incorporate missingness masks, which are binary indicators of whether a measurement is observed at a given $(s, t)$, where $s \in \mathcal{S}, t \in \mathcal{T}$. For each time step $t \in \mathcal{T}$, we construct input data as follows:

*Noisy Target Data:* The target data at a particular time step $t \in \mathcal{T}$ and location $s^* \in \mathbb{R}^d$ is represented by the noisy sample $\mathbf{x}_k^* \in \mathbb{R}^C$, initialized as Gaussian noise at inference time. We concatenate $\mathbf{x}_k^*$ with the target location coordinates $s^*$ and the corresponding missingness mask $\mathbf{m}^* \in \{0, 1\}^C$.

*Conditional Observed Data:* For each observed location $s \in \mathcal{S}$ and particular time step $t \in \mathcal{T}$, we concatenate the observed measurement $\mathbf{x} \in \mathbb{R}^C$, the location coordinates $s \in \mathbb{R}^d$, and the corresponding missingness mask $\mathbf{m} \in \{0, 1\}^C$.

The Spatial Encoder treats time steps as independent and identically distributed (IID) samples, allowing parallel computation across $\mathcal{T}$. The concatenated feature vectors for the noisy target and observed data are processed by a cross-attention mechanism, where the noisy target data serve as the query, and the observed sensor data serve as the key and value. This cross-attention computes attention weights that reflect the spatial relevance of each observed sensor to the target location, similar to the covariance-based weighting in kriging. The learnable weights within this module depend only on the combined size of feature vectors and location embedding rather than the number of locations, enabling the encoder to seamlessly handle a variable number of sensor locations. The spatial attention layers iteratively refine the representation, progressively integrating spatial information from neighboring sensors into the target predictions. We chose not to jointly model the spatial-temporal correlations to reduce the space, time, and sample complexities.

## 4.3 TEMPORAL ENCODER

Starting from the spatially contextualized target representation produced by the Spatial Encoder, the model refines it by capturing temporal and cross-feature correlations. The Temporal Encoder uses DiT (Peebles & Xie, 2023) self-attention to model temporal dependencies. We compute the self-attention on the target location only and treat the time dimension $\mathcal{T}$ as the sequence dimension.

Inputs are $X_k^* \in \mathbb{R}^{T \times C}$ for the target $s^*$ and the conditional information for the observed locations. The diffusion step $k$ is embedded with sinusoidal encodings and added to token embeddings to condition on the noise level. Self-attention operates along time (after temporal positional encodings), yielding a $T \times T$ attention map per location that captures temporal patterns across all features simultaneously. The number of learnable parameters for each layer is the sum of the feature dimension and the size of location embeddings. We concatenate the latter to inject spatial context.

## 4.4 FEATURE ENCODER

Each Feature Encoder layer uses DiT blocks with self-attention. It models correlations among $C$ features and injects spatial coordinate embeddings for spatial structure. We do this for the target location only. The feature dimension is treated as the sequence dimension. Parameters scale with the temporal dimension; conditional observed data provide context. The layer takes the output of the preceding Temporal Encoder. Self-attention computes feature-wise weights across all times at each location, learning dependencies between channels.

Temporal and Feature Encoders alternate–[Temporal $\rightarrow$ Feature]–repeated $N'_{layers}$ times (a depth hyperparameter). This interleaving jointly models temporal-feature dependencies across levels. After the final Feature Encoder, we apply the DiT output head (Peebles & Xie, 2023) (LayerNorm + linear) to the target location to predict the noise.

## 4.5 FOURIER COMPRESSION (TREND + SEASONALITY) FOR DYNASTI

To mitigate DynaSTI's slow inference on long sequences, we compress each multivariate time-series (length $L$) into a compact trend + seasonality representation in the frequency domain as a truncated Fourier series (to lowest $F$ frequency pairs). For feature $k$ with centered time $\tau_t \in [-1, 1]$, we reconstruct:

$$\hat{y}_{t,k} = \underbrace{c_k + m_k \tau_t}_{\text{trend}} + \underbrace{\sum_{f=1}^{F} \left[ \alpha_{k,f} \cos\left(\frac{2\pi ft}{L}\right) + \beta_{k,f} \sin\left(\frac{2\pi ft}{L}\right) \right]}_{\text{seasonality}}, \tag{2}$$

where the zero-th frequency component is absorbed by $c_k$. The *compressed vector* per feature is

$$\mathbf{z}_k = \left[ \alpha_{k,1:F} , \beta_{k,1:F} , c_k , m_k \right] \in \mathbb{R}^{2F+2}, \tag{3}$$

so a batch $y \in \mathbb{R}^{B \times L \times K}$ becomes $Z \in \mathbb{R}^{B \times (2F+2) \times K}$. In practice $2F+2 \ll L$, which yields faster diffusion steps despite a small coefficient-fitting overhead.

The diffusion model is trained directly on the compressed data. At each epoch, we map the observed conditionals to $(\alpha, \beta, c, m)$ by minimizing a reconstruction loss (a few gradient steps), then concatenate these coefficients to form $Z$. At inference, we apply the same mapping before running DynaSTI, reducing end-to-end latency relative to operating at length $L$. We initialize $(c, m)$ via least squares on $(\tau_t, y_{.,k})$, and set $(\alpha, \beta)$ by a one-shot real FFT of the series; all coefficients are subsequently refined by gradient descent. We are calling this model *FDynaSTI*.

## 4.6 TRAINING & INFERENCE

**Training:** For each dataset, we partition the sensor locations into non-overlapping training and testing pools, with the training pool comprising 80% of the total locations. To ensure our model, DynaSTI, effectively handles incomplete data, we preserve any naturally occurring missing values in the training data. For datasets with minimal missing data, we introduce artificial missingness by randomly masking a subset of observations at the observed locations, enabling the model to learn to handle real-world scenarios with missing sensor data.

We train our model, DynaSTI, within the Denoising Diffusion Probabilistic Model (DDPM) framework, tailored for conditional generation of spatio-temporal data. At each training step, we uniformly sample a target location $s^* \in \mathcal{S}'$, where $\mathcal{S}' \subset \mathcal{S}$ is the training pool, with its true time-series data $X_0^* \in \mathbb{R}^{T \times C}$ ($T$ time steps, $C$ features). The conditional observed data $\mathcal{D}$ comprises time-series data $X_0 \in \mathbb{R}^{T \times C}$, and missingness masks $M \in \{0,1\}^{T \times C}$ for a set of observed

Table 2: Datasets Description

| Dataset | Sampling interval | Time-series length | Number of features | Training locations | Testing locations |
|---|---|---|---|---|---|
| AWN | 15 minutes | 288 (3 days) | 7 | 54 | 13 |
| NACSE | 1 day | 30 days | 2 | 143 | 36 |
| METR-LA | 5 minutes | 288 (1 day) | 1 | 165 | 42 |
| PEMS-BAY | 5 minutes | 288 (1 day) | 1 | 260 | 65 |

locations $\mathcal{S}_{\text{obs}} \subseteq \mathcal{S}' \setminus s^*$. We sample a diffusion step $k \sim \text{Uniform}(\{1, \ldots, K\})$ and noise $\varepsilon \sim \mathcal{N}(0, \mathbf{I})$, computing the noisy target data $X_k^*$ according to Section 4.1. The model predicts the noise $\hat{\varepsilon} = \varepsilon_\theta(X_k^*, k, s^*, \mathcal{D})$, minimizing the loss in Eq. 1.

We explore two approaches to select $\mathcal{S}_{\text{obs}}$: (1) using all locations in $\mathcal{S}' \setminus s^*$ as observed sensors in each epoch; (2) randomly sampling a variable-sized subset of $\mathcal{S}' \setminus s^*$ per epoch, simulating dynamic sensor availability. Experiments show that both approaches yield comparable performance, but we adopt the first approach for training because of its simplicity throughout the paper.

**Inference:** During inference, the model starts with Gaussian noise $X_K^* \in \mathbb{R}^{T \times C} \sim \mathcal{N}(0, \mathbf{I})$ at the target location $s^*$ and iteratively applies the reverse diffusion process conditioned on the target location $s^*$, diffusion step $k$, and conditional observed data along with locations $\mathcal{D}$:

$$X_{k-1}^* \sim p_\theta(X_{k-1}^* \mid X_k^*, k, s^*, \mathcal{D}), \quad k = K, K-1, \ldots, 1.$$

In each diffusion step $k$, we get the estimated noise $\hat{\varepsilon} = \varepsilon_\theta(X_k^*, k, s^*, \mathcal{D})$ and calculate the posterior mean and variance of the $(k-1)$-th step noisy target data $\mu_{k-1} = \frac{1}{\sqrt{\alpha_k}}(X_k^* - \frac{\beta_k}{\sqrt{1-\bar{\alpha}_k}}\hat{\varepsilon})$ and $\sigma_{k-1} = \frac{1-\bar{\alpha}_{k-1}}{1-\bar{\alpha}_k}\beta_k$ respectively. Then, we get $X_{k-1}^* = \mathcal{N}(\mu_{k-1}, \sigma_{k-1}\mathbf{I})$ and repeat this procedure, decrementing $k$ until $k = 1$, at which point we recover our prediction for the target location $X_0^*$.

## 5 EVALUATION

In this section, we evaluate the performance of our proposed diffusion-based generative model for multivariate spatio-temporal data interpolation. We conduct experiments on four real-world datasets and compare our model against the following baseline methods based on their availability and handling multivariate data: (1) Deterministic methods: Mean imputation, DeepKriging (Nag et al., 2023), KITS (Li et al., 2023), GSLI (Yang et al., 2025), and IGNNK (Wu et al., 2021), (2) Probabilistic methods: ST-VGP (Hamelijnck et al., 2021), ImputeFormer (Nie et al., 2024), and PriSTI (Liu et al., 2023). Additionally, we perform an ablation study to assess the contribution of each component in our architecture.

### 5.1 EXPERIMENTAL SETUP

We utilize four diverse spatio-temporal datasets to evaluate our model. **AWN** dataset consists of weather data collected from the AgWeatherNet network [1] such as temperatures at two different pressure levels, relative humidity, dewpoint, wind speed, wind gust, and solar radiation. **NACSE** dataset provides daily maximum and minimum temperature data from 179 weather stations in Northwest Oregon, sourced from the NACSE PRISM climate dataset [2]. **METR-LA** and **PEMS-BAY** traffic datasets contain traffic speed data collected at 5-minute intervals from Los Angeles and San Francisco Bay Area (He, 2025) respectively. Table 2 shows the attributes of the four datasets.

Across datasets, we partition locations into disjoint train/test pools. In each trial, we sample a test location $s^*$, input its coordinates along with training-pool's locations and observations $\mathcal{D}$, and use $s^*$'s multivariate time-series as ground truth. We enforce a strict temporal split—train on earlier data, test on the final 20%—for all methods except ST-VGP (Hamelijnck et al., 2021). As a Gaussian Process method, ST-VGP's kernel trained on earlier windows extrapolates poorly under temporal shift, so a disjoint split would unfairly penalize it. We therefore fit ST-VGP on observations from the evaluation period–matching the test window of other methods–and predict the held-out targets,

---

[1] https://weather.wsu.edu/
[2] https://shorturl.at/Aor04

Table 3: Comparison of the RMSE ($\pm$ 95% confidence interval) across four datasets

| Model | NACSE | AWN | METR-LA | PEMS-BAY |
|---|---|---|---|---|
| MEAN | $0.4113 \pm 0.0590$ | $0.9826 \pm 0.0117$ | $1.4954 \pm 0.0269$ | $0.9927 \pm 0.0646$ |
| DeepKriging | $0.8965 \pm 0.0777$ | $0.5651 \pm 0.0142$ | $1.3310 \pm 0.0555$ | $0.9524 \pm 0.0842$ |
| ST-VGP | $0.6408 \pm 0.0133$ | $0.5121 \pm 0.0183$ | $1.1577 \pm 0.0442$ | $0.8990 \pm 0.0333$ |
| KITS | $0.4009 \pm 0.0251$ | $0.4453 \pm 0.0171$ | $1.1502 \pm 0.0425$ | $0.8542 \pm 0.0391$ |
| GSLI | $0.6215 \pm 0.0125$ | $0.4567 \pm 0.0145$ | $1.1521 \pm 0.0450$ | $0.8406 \pm 0.0534$ |
| IGNNK | $0.8542 \pm 0.0224$ | $0.5400 \pm 0.0139$ | $1.3217 \pm 0.0538$ | $0.9446 \pm 0.0986$ |
| PriSTI | $0.5904 \pm 0.0574$ | $0.4766 \pm 0.0171$ | $1.1824 \pm 0.0553$ | $0.8762 \pm 0.0930$ |
| ImputeFormer | $0.3521 \pm 0.0777$ | $0.4436 \pm 0.0181$ | $1.1433 \pm 0.0725$ | $0.8432 \pm 0.0629$ |
| DynaSTI | $\mathbf{0.2333 \pm 0.0760}$ | $0.4339 \pm 0.0163$ | $1.1216 \pm 0.0687$ | $0.8252 \pm 0.0963$ |
| FDynaSTI | $0.2608 \pm 0.0597$ | $\mathbf{0.3706 \pm 0.0186}$ | $\mathbf{1.0338 \pm 0.0338}$ | $\mathbf{0.7996 \pm 0.0522}$ |

Table 4: Comparison of the CRPS and MIS ($\pm$ 95% confidence interval) across four datasets

| Model | NACSE | | AWN | | METR-LA | | PEMS-BAY | |
|---|---|---|---|---|---|---|---|---|
| | CRPS | MIS | CRPS | MIS | CRPS | MIS | CRPS | MIS |
| ST-VGP | $0.2351 \pm 0.0521$ | $15.21 \pm 0.67$ | $0.3211 \pm 0.0230$ | $7.32 \pm 0.07$ | $0.8532 \pm 0.3320$ | $10.78 \pm 0.20$ | $0.8245 \pm 0.0341$ | $15.65 \pm 0.42$ |
| PriSTI | $0.2740 \pm 0.0477$ | $23.46 \pm 0.94$ | $0.3542 \pm 0.0121$ | $12.71 \pm 0.23$ | $0.9777 \pm 0.0355$ | $13.43 \pm 0.03$ | $0.9573 \pm 0.0886$ | $19.18 \pm 0.02$ |
| DynaSTI | $\mathbf{0.1631 \pm 0.0515}$ | $\mathbf{11.17 \pm 0.86}$ | $0.2790 \pm 0.0098$ | $4.37 \pm 0.15$ | $0.6839 \pm 0.0350$ | $6.85 \pm 0.46$ | $\mathbf{0.6392 \pm 0.0308}$ | $9.12 \pm 0.72$ |
| FDynaSTI | $0.1933 \pm 0.0436$ | $13.22 \pm 0.87$ | $\mathbf{0.2440 \pm 0.0110}$ | $\mathbf{2.17 \pm 0.11}$ | $\mathbf{0.6528 \pm 0.0049}$ | $\mathbf{5.58 \pm 0.59}$ | $0.6832 \pm 0.0361$ | $\mathbf{7.74 \pm 0.80}$ |

a setup that favors ST-VGP. For each experiment, we run 10 trials per time period, varying the target location $s^*$ to assess robustness across different spatial configurations.

We evaluate using RMSE at target locations, CRPS, and MIS. All metrics are reported as the mean with 95% confidence intervals over 10 runs. CRPS and MIS are probabilistic scoring rules (lower is better) that assess uncertainty quality—rewarding calibrated, sharp distributions and penalizing over- or under-confident predictions.

## 5.2 RESULTS

We evaluate our model's performance across diverse scenarios, including virtual sensors and dynamic sensor configurations. In addition, we conducted an ablation study to evaluate the contribution of each of the key architectural components. Given that features vary in scale, we calculate the RMSE, CRPS, and MIS using normalized predictions and ground truth values.

Across all four datasets, our approach attains the best RMSE in Table 3. The Fourier variant (FDynaSTI) outperforms DynaSTI in all datasets except NACSE, which has relatively short time series. Relative to the strongest baseline model per dataset, the error reductions are 16.8% (AWN against KITS), 41.8% (NACSE against KITS), 10.1% (METR-LA against KITS), and 4.9% (PEMS-BAY against GSLI). These gains hold within the reported 95% confidence intervals.

DynaSTI/FDynaSTI achieves the lowest CRPS and MIS on all datasets in Table 4, indicating better-calibrated and sharper predictive distributions. For CRPS, FDynaSTI closely matches DynaSTI on METR-LA (0.6528 vs. 0.6839) and outperforms all on AWN, while DynaSTI leads on PEMS-BAY and NACSE (lower CRPS is better). In terms of MIS, FDynaSTI achieves the best performance across all datasets except NACSE, for which DynaSTI performs best. Note that deterministic baselines are omitted since CRPS and MIS does not apply to them.

To test the impact of dynamic sensor failures or sensor dropouts during inference, we gradually masked the input sensors randomly at inference time and show the results in Table 5 and performance degrades gracefully. From 100% to 10% active sensors, RMSE increases are +14.2% (AWN), +61.7% (NACSE), +85.6% (METR-LA), and +24.3% (PEMS-BAY), reflecting dataset difficulty while preserving competitive accuracy under severe sparsity.

Table 6 presents the results of an ablation study, where we remove key components of our model–Spatial Encoder (SE), Temporal Encoder (TE), and Feature Encoder (FE)–to assess their individual contributions. The Spatial Encoder is critical for maintaining model performance. Removing it more than doubles error on AWN and yields large degradations on NACSE, METR-LA, and PEMS-BAY. Removing the Temporal Encoder hurts notably on NACSE and METR-LA. The Feature Encoder

Table 5: Evaluation of our model's performance (RMSE $\pm$ 95% confidence interval) under varying percentages of active input sensors during inference

| Dataset | Percentage of Active Input Sensors | | | | | |
|---|---|---|---|---|---|---|
| | 100% | 90% | 70% | 50% | 30% | 10% |
| AWN | $0.3706 \pm 0.0186$ | $0.3770 \pm 0.0195$ | $0.3872 \pm 0.0303$ | $0.3860 \pm 0.0173$ | $0.4101 \pm 0.0189$ | $0.4231 \pm 0.0190$ |
| NACSE | $0.2333 \pm 0.0760$ | $0.2560 \pm 0.0931$ | $0.2868 \pm 0.0995$ | $0.3291 \pm 0.0987$ | $0.3302 \pm 0.1021$ | $0.3772 \pm 0.1102$ |
| METR-LA | $1.0338 \pm 0.0338$ | $1.1922 \pm 0.0525$ | $1.3132 \pm 0.0566$ | $1.5083 \pm 0.0474$ | $1.8231 \pm 0.1607$ | $1.9191 \pm 0.1404$ |
| PEMS-BAY | $0.7996 \pm 0.0522$ | $0.8292 \pm 0.0942$ | $0.8878 \pm 0.0926$ | $0.9232 \pm 0.0974$ | $0.9421 \pm 0.0912$ | $0.9938 \pm 0.0983$ |

Table 6: Ablation study on the four datasets. Each cell reports RMSE ($\pm$ 95% confidence interval). SE: Spatial Encoder, TE: Temporal Encoder, FE: Feature Encoder.

| Dataset | DynaSTI/FDynaSTI | no SE | no TE | no FE | no TE & no FE |
|---|---|---|---|---|---|
| AWN | $\mathbf{0.3706 \pm 0.0186}$ | $0.7836 \pm 0.0067$ | $0.3737 \pm 0.0171$ | $0.4192 \pm 0.0069$ | $0.4209 \pm 0.0049$ |
| NACSE | $\mathbf{0.2333 \pm 0.0760}$ | $0.4462 \pm 0.0749$ | $0.3512 \pm 0.0726$ | $0.3376 \pm 0.0784$ | $0.3628 \pm 0.0779$ |
| METR-LA | $\mathbf{1.0338 \pm 0.0338}$ | $1.7701 \pm 0.0788$ | $1.4086 \pm 0.0774$ | N/A | N/A |
| PEMS-BAY | $\mathbf{0.7996 \pm 0.0522}$ | $1.1156 \pm 0.1036$ | $1.0124 \pm 0.0417$ | N/A | N/A |

matters for multivariate datasets (AWN and NACSE), while it is inapplicable to the two univariate traffic datasets. For both Table 5 and Table 6, we reported the RMSE values for the best performing model in Table 3, i.e., FDynaSTI for AWN, METR-LA, and PEMS-BAY, and DynaSTI for NACSE.

Table 7: Inference time (seconds) for each model to predict the entire time series at a single virtual sensor location on each of the four datasets.

| Dataset | DynaSTI | FDynaSTI | IGNNK | ST-VGP | DeepKriging | PriSTI | KITS | GSLI |
|---|---|---|---|---|---|---|---|---|
| AWN | 168.025s | 28.920s | 0.003s | 2.211s | 1.224s | 59.708s | 0.003 | 0.004s |
| NACSE | 59.264s | 39.357s | 0.001s | 0.025s | 0.419s | 26.603s | 0.001 | 0.001s |
| METR-LA | 200.374s | 24.880s | 0.002s | 3.231s | 1.445s | 33.999s | 0.002 | 0.002s |
| PEMS-BAY | 292.911s | 32.466s | 0.003s | 3.570s | 1.256s | 52.151s | 0.003 | 0.003s |

Table 7 shows the inference speeds of different models. Not surprisingly, the deterministic models are much faster than the generative models. FDynaSTI is significantly faster than DynaSTI and the other diffusion model, PriSTI. However it is slower than the other generative model, ST-VGP. We experimented with replacing the DDPM design of DynaSTI with DDIM (Song et al., 2020); however, we found that it significantly degrades the quality of the generated samples. All experiments were conducted on a cluster using GPU-enabled nodes equipped with Nvidia Tesla v100 32GB GPUs.

# 6 DISCUSSION AND CONCLUSIONS

Across four real-world datasets, DynaSTI delivers the best point accuracy (lowest RMSE, Table 3) and the best probabilistic quality (lowest CRPS, Table 4), showing that conditioning a diffusion model on spatial, temporal, and feature context yields both accurate means and well-calibrated, sharp uncertainty. The Fourier-compressed variant (FDynaSTI) outperforms or matches DynaSTI on several settings, offering faster sampling. Moreover, initializing harmonic coefficients with real FFT substantially improves accuracy over random starts (Appendix A.3, Table 9). DynaSTI is robust to time-varying sensor availability. As active sensors drop from 100% to 10%, errors rise gradually but remain competitive across datasets (Table 5). Ablations confirm the Spatial Encoder is critical (its removal more than doubles RMSE on AWN and NACSE), while Temporal and Feature encoders provide complementary gains (Table 6).

In summary, our proposed model significantly advances spatio-temporal interpolation by delivering a flexible, efficient, and accurate solution specifically designed to handle the complexities of real-world sensor networks. Its ability to manage missing data, adapt to changing configurations, and provide probabilistic predictions makes it a useful tool for applications such as environmental monitoring and traffic analysis. Moreover, as a generative model, it opens avenues for generating synthetic data, which could be valuable for training other models or conducting simulations in data-scarce environments.

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

## A APPENDIX

### A.1 LLM USAGE

We used ChatGPT to aid in writing. We used it only to shorten sentences and improve grammar.

### A.2 INCOMPLETENESS OF DATASETS

The four datasets we used have some original missing values. Table 8 shows the percentage of missing values inherent to them.

Table 8: Percentage of missingness in the four datasets

| Dataset | Percentage of missingness (%) |
|---------|-------------------------------|
| NACSE | 24.76 |
| AWN | 53.56 |
| METR-LA | 8.11 |
| PEMS-BAY | 0.003 |

## A.3 RANDOM VS REAL FFT INITIALIZATION OF FOURIER COEFFICIENTS

Empirically, random initialization of the seasonal (sine/cosine) coefficients performs substantially worse than *real FFT (rFFT)* initialization. Random coefficients correspond to an arbitrary spectrum that poorly reconstructs the series in the time domain at the start of training, forcing gradient descent to discover both amplitudes and phases from scratch. In contrast, rFFT initialization aligns the initial parameters with the dominant spectral content of the detrended signal, placing optimization near a good basin. As a result, rFFT initialization achieves lower error with far fewer epochs, whereas random initialization typically requires many more updates to reach a comparable minimum—undermining the intended end-to-end speedup of our Fourier-compressed diffusion pipeline. For a fair comparison and to preserve the diffusion model's overall speedup, the Fourier model under both initialization schemes was trained for the same number of epochs; results are shown in Table 9.

Table 9: FDynaSTI's Fourier transform random vs real FFT initialization

| Dataset | FDynaSTI-random | FDynaSTI-rFFT |
|---------|-----------------|---------------|
| AWN | $0.9889 \pm 0.0071$ | $\mathbf{0.3706 \pm 0.0186}$ |
| NACSE | $0.5651 \pm 0.0621$ | $\mathbf{0.2608 \pm 0.0597}$ |
| METR-LA | $1.1872 \pm 0.0444$ | $\mathbf{1.0338 \pm 0.0338}$ |
| PEMS-BAY | $1.0481 \pm 0.1094$ | $\mathbf{0.7996 \pm 0.0522}$ |

## A.4 HYPERPARAMETERS

Tables 10, 11, 12, 13, 14, 15, and 16 record the hyperparameters used for the corresponding model and dataset. All hyperparameters were tuned via grid search over predefined candidate values.

Table 10: DynaSTI/FDynaSTI hyperparameters

| Hyperparameter | NACSE | AWN | METR-LA | PEMS-BAY |
|----------------|-------|-----|---------|----------|
| Epoch | 2000/800 | 1000/600 | 1000/600 | 1000/600 |
| Lr | 1.0e-4 | 1.0e-4 | 1.0e-3 | 1.0e-4 |
| $\beta_{start}$ | 0.0001 | 0.0001 | 0.0001 | 0.0001 |
| $\beta_{end}$ | 0.1 | 0.1 | 0.1 | 0.1 |
| Diffusion steps | 50 | 50 | 50 | 50 |
| Spatial context embedding | 128 | 128 | 128 | 256 |
| Spatial encoder layers | 4 | 4 | 4 | 4 |
| Temporal & Feature encoder layers | 4 | 4 | 4 | 4 |
| FDynaSTI Fourier transform parameters | | | | |
| iterations | 200 | 100 | 100 | 100 |
| Lr | 0.001 | 0.01 | 0.01 | 0.01 |
| F | 7 | 16 | 16 | 16 |

Table 11: ST-VGP hyperparameters

| Hyperparameter | NACSE | AWN | METR-LA | PEMS-BAY |
|---|---|---|---|---|
| Likelihood noise | 2.0 | 2.0 | 1.0 | 1.0 |
| Variance | 1.0 | 1.0 | 0.5 | 1.0 |
| Lengthscale | (0.001,0.1,0,1) | (0.001,0.2,0,2) | (0.01,0.3,0,3) | (0.01,0.2,0,2) |
| Lr Adam | 0.001 | 0.001 | 0.001 | 0.001 |
| Lr Newton | 0.1 | 0.1 | 0.1 | 0.1 |
| Epoch | 300 | 500 | 500 | 600 |

Table 12: PriSTI hyperparameters

| Hyperparameter | NACSE | AWN | METR-LA | PEMS-BAY |
|---|---|---|---|---|
| Epoch | 2000 | 1000 | 1000 | 1000 |
| Lr | 1.0e-4 | 1.0e-4 | 1.0e-3 | 1.0e-4 |
| $\beta_{start}$ | 0.0001 | 0.0001 | 0.0001 | 0.0001 |
| $\beta_{end}$ | 0.1 | 0.1 | 0.1 | 0.1 |
| Diffusion steps | 50 | 50 | 50 | 50 |
| Layers | 4 | 4 | 4 | 4 |
| Channels | 64 | 32 | 64 | 64 |
| Number of heads | 8 | 8 | 8 | 8 |
| Projection dim | 16 | 16 | 16 | 16 |
| Time embedding dim | 128 | 128 | 128 | 128 |
| Feature embedding dim | 16 | 16 | 16 | 16 |

Table 13: KITS hyperparameters

| Hyperparameter | NACSE | AWN | METR-LA | PEMS-BAY |
|---|---|---|---|---|
| Epoch | 300 | 500 | 300 | 300 |
| Lr | 0.001 | 0.0001 | 0.001 | 0.001 |
| Samples per epoch | 5120 | 5120 | 5120 | 5120 |
| Hidden layer dim | 64 | 64 | 64 | 64 |

Table 14: GSLI hyperparameters

| Hyperparameter | NACSE | AWN | METR-LA | PEMS-BAY |
|---|---|---|---|---|
| Epoch | 100 | 200 | 200 | 300 |
| Lr | 0.001 | 0.001 | 0.001 | 0.001 |
| Channels | 128 | 64 | 64 | 64 |
| Projection dim | 128 | 64 | 64 | 64 |
| Time embedding dim | 128 | 128 | 128 | 128 |
| Feature embedding dim | 16 | 16 | 16 | 16 |
| Number of heads | 8 | 8 | 8 | 8 |

Table 15: IGNNK hyperparameters

| Hyperparameter | NACSE | AWN | METR-LA | PEMS-BAY |
|---|---|---|---|---|
| Epoch | 5000 | 2000 | 2000 | 3000 |
| Lr | 1.0e-4 | 1.0e-5 | 1.0e-4 | 1.0e-6 |
| Embedding dim | 128 | 256 | 512 | 256 |
| Order | 1 | 3 | 3 | 3 |

Table 16: DeepKriging hyperparameters

| Hyperparameter | NACSE | AWN | METR-LA | PEMS-BAY |
|---|---|---|---|---|
| Epoch | 500 | 600 | 700 | 700 |
| Lr | 1.0e-3 | 1.0e-3 | 1.e-4 | 1.0e-4 |

## A.5 QUALITATIVE RESULTS

Figures 2 and 3 visualize interpolation at two NACSE stations held out as virtual targets. The red curve is ground truth; purple and blue are the posterior means from DynaSTI and FDynaSTI. Shaded bands (pink for DynaSTI, cyan for FDynaSTI) denote the $\pm 3\sigma$ envelopes from generated samples, quantifying predictive uncertainty. Each figure also includes a map of Spatial Encoder attention over observed stations, indicating which sensors contribute most to the target prediction. Blue represents more attention-weight and green represents less.

Figures 4 and 5 show results for two stations in the AWN dataset showing the results of DynaSTI and FDynaSTI for all seven features.

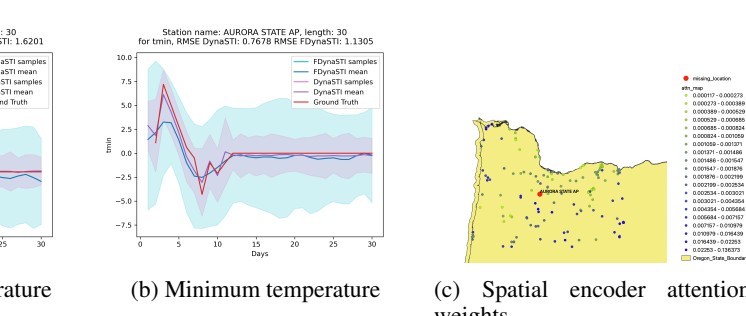

(a) Maximum temperature

(b) Minimum temperature

(c) Spatial encoder attention weights

Figure 2: NACSE dataset, missing station "Aurora State AP"

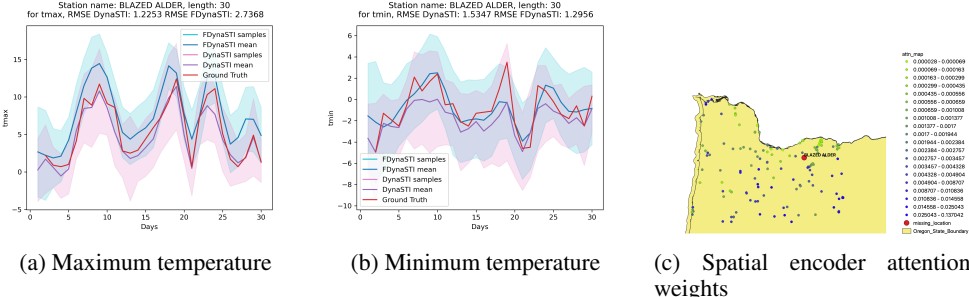

(a) Maximum temperature

(b) Minimum temperature

(c) Spatial encoder attention weights

Figure 3: NACSE dataset, missing station "Blazed Alder"

## A.6 FDYNASTI FOURIER COEFFICIENT FITTING TIME

Table 17: Fourier coefficient fitting wall-clock time in seconds

| Dataset | time (s) |
|---|---|
| NACSE | 0.097 |
| AWN | 0.059 |
| METR-LA | 0.083 |
| PEMS-BAY | 0.071 |

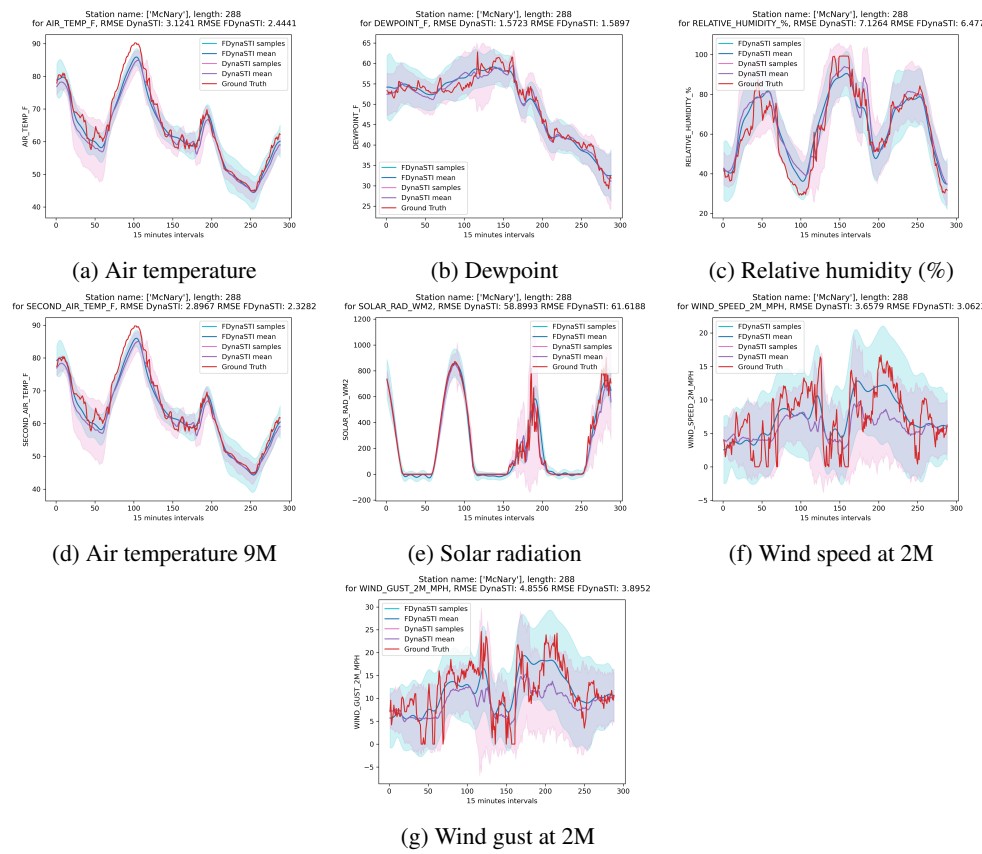

(a) Air temperature      (b) Dewpoint      (c) Relative humidity (%)

(d) Air temperature 9M      (e) Solar radiation      (f) Wind speed at 2M

(g) Wind gust at 2M

Figure 4: AWN dataset, missing station "McNary"

## A.7 TRAINING TIME COMPARISON

Table 18 shows the training time comparison of all the models on all four datasets.

Table 18: Training time comparison of the models on all four datasets in d-hh:mm:ss format

| Model | NACSE | AWN | METR-LA | PEMS-BAY |
|---|---|---|---|---|
| DeepKriging | 00:20:12 | 01:08:31 | 01:58:09 | 02:07:57 |
| ST-VGP | 00:18:02 | 00:51:32 | 01:03:56 | 01:46:47 |
| KITS | 01:53:45 | 02:42:28 | 03:23:19 | 03:37:17 |
| GSLI | 00:15:10 | 00:37:42 | 01:10:23 | 01:22:34 |
| IGNNK | 00:21:47 | 01:17:13 | 02:12:30 | 02:19:14 |
| PriSTI | 00:58:52 | 03:59:15 | 06:32:38 | 06:56:29 |
| ImputeFormer | 00:26:21 | 01:21:42 | 02:10:24 | 02:18:57 |
| DynaSTI | 01:01:20 | 03:54:25 | 06:29:10 | 06:40:23 |
| FDynaSTI | 00:35:56 | 01:48:21 | 02:42:17 | 02:57:58 |

## A.8 SENSITIVITY OF F TO LOSS AND INFERENCE TIME

Figure 6 illustrates the sensitivity of FDynaSTI's performance to different values of F, while Figure 7 reports the corresponding inference times across all four datasets. As shown, increasing F generally reduces the loss but also leads to higher inference time. Therefore, as summarized in Table 10, we selected the final value of F by balancing this trade-off between accuracy and computational cost.

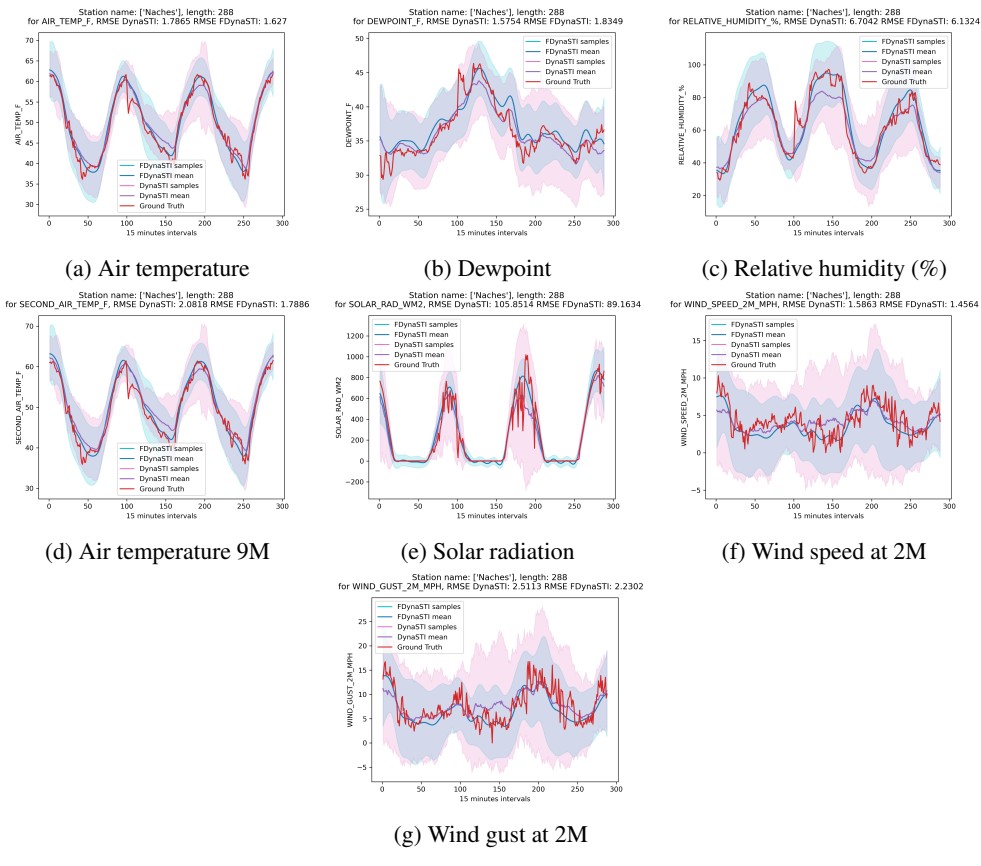

(a) Air temperature      (b) Dewpoint      (c) Relative humidity (%)

(d) Air temperature 9M      (e) Solar radiation      (f) Wind speed at 2M

(g) Wind gust at 2M

Figure 5: AWN dataset, missing station "Naches"

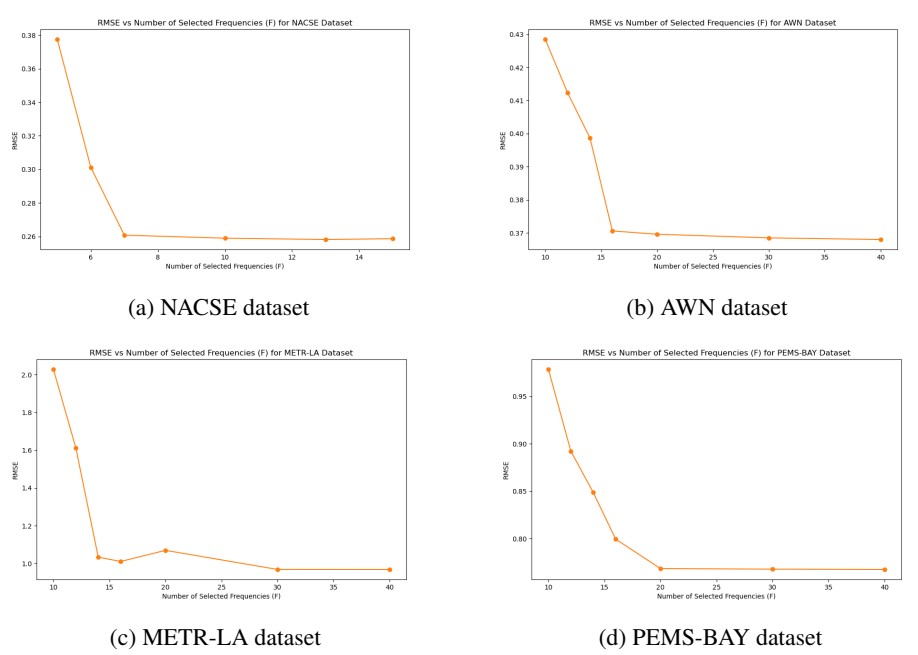

(a) NACSE dataset      (b) AWN dataset

(c) METR-LA dataset      (d) PEMS-BAY dataset

Figure 6: F vs RMSE of the FDynaSTI model for all four datasets

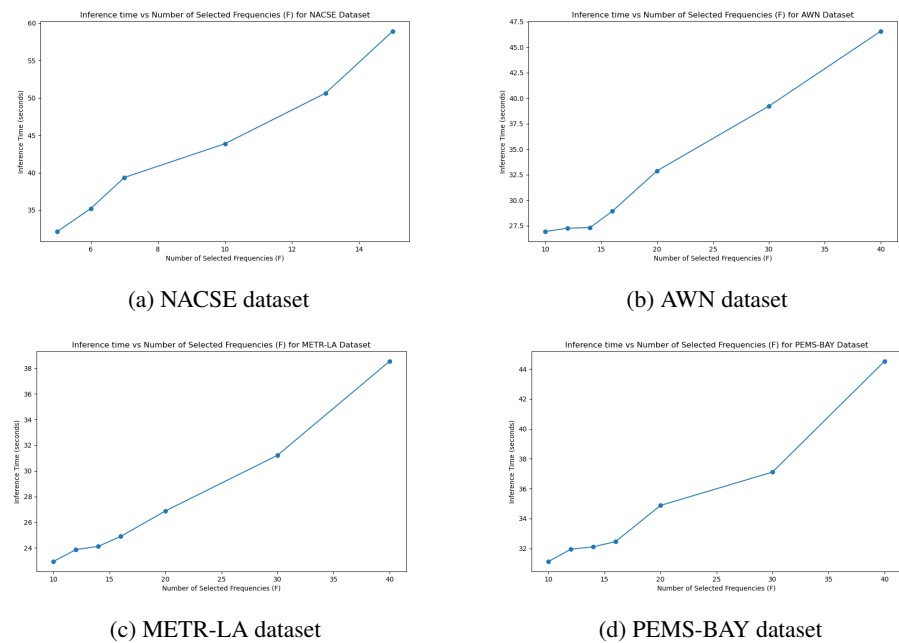

Figure 7: F vs inference time of the FDynaSTI model for all four datasets

## A.9 DATASET VISUALIZATION

The NACSE PRISM Climate dataset contains daily maximum and minimum temperatures from 176 weather stations in Northeast Oregon, recorded at 15-minute intervals from January 2011 to December 2021. For testing, we use the final two years of data.

The AgWeatherNet (AWN) dataset provides weather observations from 67 stations in Southeast Washington, spanning the period from 2007 to 2023. Each weather station contains features such as temperatures at two different pressure levels, relative humidity, dewpoint, wind speed, wind gust, and solar radiation.

The METR-LA dataset includes traffic speed measurements collected from 207 loop detectors installed across Los Angeles County highways. It covers a four-month period from March 1, 2012, to June 30, 2012.

The PeMS-Bay dataset, collected by Caltrans' Performance Measurement System (PeMS), contains traffic speed data from 325 sensors across the Bay Area. It covers six months of data from January 1 to May 31, 2017.

For all datasets, training and testing stations were randomly selected to form our experimental splits. Figure 8 illustrates the spatial distribution of the training and testing locations for each dataset.

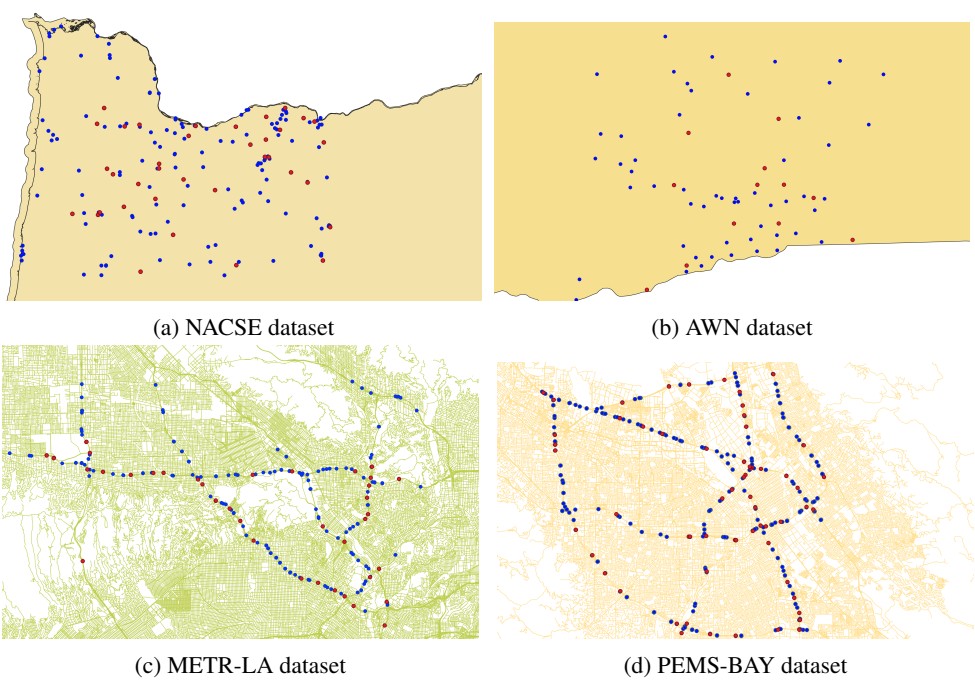

(a) NACSE dataset

(b) AWN dataset

(c) METR-LA dataset

(d) PEMS-BAY dataset

Figure 8: Training and testing location topology of all four datasets. The blue dots indicate training locations/sensors, and the red dots indicate testing locations/sensors.

