# OpenReview forum: "Diffusion-based Spatio-temporal Interpolation with Dynamic Sensor Sets"
_ICLR.cc/2026/Conference — Submitted to ICLR 2026_

### Official Review · Reviewer_wtC9 · 2025-10-17

**Soundness:** 2
**Presentation:** 2
**Contribution:** 2
**Rating:** 2
**Confidence:** 3

**Summary:**

This paper presents DynaSTI, a diffusion-based framework for spatio-temporal imputation that models uncertainty under dynamic sensor topologies. The method integrates spatial cross-attention, temporal and feature DiT encoders, and introduces a frequency-domain variant (FDynaSTI) for improved runtime efficiency.

**Strengths:**

The framework is modular and conceptually well-motivated, combining diffusion with attention for flexible inductive inference. The frequency-domain variant (FDynaSTI) effectively demonstrates the potential of harmonic compression for long time series.

**Weaknesses:**

- Regarding the method, while FDynaSTI improves runtime through frequency-domain compression, the overall framework integrates diffusion (iterative denoising) with full attention modules, which likely incurs substantial computational overhead. The paper reports only performance metrics without any inference time comparison table. It is necessary to clarify the adopted sampling strategy (e.g., DDIM), specify the denoising step number K, and include a performance-versus-diffusion-step analysis.
- Regarding related work, several closely relevant spatio-temporal interpolation models [1–2] are missing from the discussion. Additionally, the citation of USTD around line 71 seems to correspond to [4] rather than [3]?
- Regarding experiments, the study does not evaluate on large-scale datasets such as [5], and strong recent baselines, including ImputeFormer, are absent from the comparison, limiting the empirical comprehensiveness.

**Questions:**

See weaknesses.

---

> ### Author Response · Authors · 2025-11-13
>
> Could you please share the references for the works you mentioned in your review? They would be very helpful for us in formulating our response.

---

> ### Author Response · Authors · 2025-11-22
>
> **Regarding the method, while FDynaSTI improves runtime through frequency-domain compression, the overall framework integrates diffusion (iterative denoising) with full attention modules, which likely incurs substantial computational overhead. The paper reports only performance metrics without any inference time comparison table. It is necessary to clarify the adopted sampling strategy (e.g., DDIM), specify the denoising step number K, and include a performance-versus-diffusion-step analysis.**
>
> Answer: We have already included the inference-time comparison with other models in Table 7. Additionally, our diffusion sampling strategy, DDPM, has already been described in the paper. As noted in lines 464–465, replacing DDPM with DDIM degrades sampling quality. This explanation was present in the original (non-revised) version of the manuscript.
>
> **Regarding related work, several closely relevant spatio-temporal interpolation models [1–2] are missing from the discussion. Additionally, the citation of USTD around line 71 seems to correspond to [4] rather than [3]?**
>
> Answer: We have corrected the USTD citation in the revised manuscript. However, we are unsure which spatio-temporal interpolation models you are referring to, as the corresponding references were not provided. Could you please clarify or share the specific references so we can address them appropriately?
>
> **Regarding experiments, the study does not evaluate on large-scale datasets such as [5], and strong recent baselines, including ImputeFormer, are absent from the comparison, limiting the empirical comprehensiveness.**
>
> Answer: We have included the ImputeFormer results in the revised manuscript. However, we are not certain which large-scale dataset you are referring to, as the relevant reference was not provided. Could you please clarify or share the specific citation so we can address it properly?

---

> ### Comment · Reviewer_wtC9 · 2025-11-24
>
> Thank you very much for your rebuttal, and apologies that I did not include the explicit references in my original review.
> The works I referred to are:
>
>
> [1] Spatio-Temporal Field Neural Networks for Air Quality Inference. IJCAI, 2024.
>
> [2] Causality-Aware Spatiotemporal Graph Neural Networks for Spatiotemporal Time Series Imputation. CIKM, 2024.
>
> [3] UniSTD: Towards Unified Spatio-Temporal Learning across Diverse Disciplines. CVPR. 2025.
>
> [4] Towards Unifying Diffusion Models for Probabilistic Spatio-Temporal Graph Learning. SIGSPATIAL. 2024.
>
> [5] LargeST: A Benchmark Dataset for Large-Scale Traffic Forecasting. NeurIPS. 2023.

---

> ### Author Response · Authors · 2025-11-26
>
> We have added [1] and [2] to the Related Works section (Lines 109–114) in the revised version. However, we are unable to experiment with large-scale datasets such as [3] due to GPU memory constraints, which limit the feasibility of training and testing models at that scale.
>
> **References:**
>
> [1] Spatio-Temporal Field Neural Networks for Air Quality Inference. IJCAI, 2024.
>
> [2] Causality-Aware Spatiotemporal Graph Neural Networks for Spatiotemporal Time Series Imputation. CIKM, 2024.
>
> [3] LargeST: A Benchmark Dataset for Large-Scale Traffic Forecasting. NeurIPS. 2023.

---

### Official Review · Reviewer_fN4f · 2025-10-27

**Soundness:** 2
**Presentation:** 2
**Contribution:** 2
**Rating:** 2
**Confidence:** 4

**Summary:**

The paper introduces DynaSTI, a diffusion-based generative framework for estimating missing or unobserved sensor data in dynamic and incomplete sensor networks. Unlike prior models that require fixed sensor configurations, DynaSTI generalizes to unseen locations without retraining and handles missing data directly through a unified conditioning strategy that integrates spatial, temporal, and feature information. A Fourier-domain variant accelerates inference by compressing time series into trend and seasonality components. Experiments on four real-world datasets show that DynaSTI achieves better accuracy and uncertainty calibration, outperforming baselines while maintaining robustness under sensor dropout and dynamic network changes.

**Strengths:**

1. The method is well-designed for dynamic sensor topologies and inductive generalization, and Fourier variant accelerates inference significantly without significant accuracy loss.

2. It shows clear SOTA results in both deterministic and probabilistic metrics with maintain accuracy under highly incomplete rates.

**Weaknesses:**

1. The paper lacks sufficient description (although table 2 shows datasets description) or visualization of the datasets, making it difficult for readers unfamiliar with those 4 real-world datasets to assess task complexity or interpretability. For example, the reason behind the split of training/testing locations, and how they separate each other or the topology examples.

2. From the problem setup, the mask is fixed locations on the spatial coordinates, but times are regularly sampled not missing and masks are fixed across whole timesteps. In contrast, other studies address scenarios where data are missing in both spatial and temporal dimensions, with missing observations varying at each timestep - a considerably more challenging setting. I am wondering whether the proposed method maintains robust under these more challenging scenarios.

**Questions:**

1. Equation (1) has typo on the parentheses.

---

> ### Author Response · Authors · 2025-11-22
>
> **The paper lacks sufficient description (although table 2 shows datasets description) or visualization of the datasets, making it difficult for readers unfamiliar with those 4 real-world datasets to assess task complexity or interpretability. For example, the reason behind the split of training/testing locations, and how they separate each other or the topology examples.**
>
> Answer: The paper reports the amount and percentage of missing data for each dataset in Appendix - A.2 - Table 8. The training and testing locations were selected randomly, and their spatial topology is visualized in Appendix - A.9, Figure - 8.
>
> **From the problem setup, the mask is fixed locations on the spatial coordinates, but times are regularly sampled not missing and masks are fixed across whole timesteps. In contrast, other studies address scenarios where data are missing in both spatial and temporal dimensions, with missing observations varying at each timestep - a considerably more challenging setting. I am wondering whether the proposed method maintains robust under these more challenging scenarios.**
>
> Answer: There seems to be a misunderstanding. For the virtual (target) sensors, the mask contains zeros across the entire time-series simply because no observations are available. However, for the input sensors, we do not impose any assumptions on the pattern of missingness. As described in Section 3.1, our framework fully supports masks with zeros at any time step, for any feature, and for any sensor. Therefore, it already accommodates the more challenging and irregular missing-data scenarios.
>
> **Equation (1) has typo on the parentheses.**
>
> Answer: We have already fixed that in our revision. Thanks for catching the typo.

---

### Official Review · Reviewer_Wog1 · 2025-10-30

**Soundness:** 3
**Presentation:** 3
**Contribution:** 3
**Rating:** 6
**Confidence:** 2

**Summary:**

The paper proposes a diffusion-model framework for spatio-temporal interpolation in settings where sensor networks are sparse, partially observed, and dynamically changing over time. The proposed method, DynaSTI can handle unseen locations and arbitrary missing sensor observations by conditioning its diffusion denoising process on available sensor observations and their spatial coordinates. The model integrates spatial, temporal, and feature encoders to capture multi-scale dependencies and introduces a Fourier-domain compression (FDynaSTI) to accelerate inference for long time sequences. Evaluated on real-world datasets, DynaSTI achieves state-of-the-art accuracy and probabilistic calibration (CRPS) while maintaining robustness under sensor dropout.

**Strengths:**

The paper addresses a practical problem with high flexibility, supporting dynamically changing sensor configurations over time, and effectively handling unseen irregular locations as well as missing observations.

The trend+seasonality representation is effective that incurs minimal coefficient-fitting overhead while enabling significantly faster sampling, with the rFFT initialization shown to be empirically useful.

DynaSTI and FDynaSTI consistently outperform all compared methods in terms of both RMSE and CRPS.

**Weaknesses:**

In this paper, "dynamic" refers to the sensor network changing over time, while the observation distribution remains stationary. The authors should clarify this distinction in the introduction.

Spatial cross-attention can be computationally expensive since the method uses all observations to predict the target location. The author has handled the temporal computation cost with trend+seasonality representation, while didn't provide a solution for spatial overhead.

In the experimental section, although the paper claims to provide probabilistic predictions, it only reports CRPS. Additional calibration metric could be added to provide a more comprehensive evaluation.

**Questions:**

Line 420 states "DynaSTI leads on AWN and NACSE," but this appears inconsistent with Table 4. Could the authors clarify which is correct?

What are the average iteration counts and wall-clock times for the Fourier coefficient fitting at inference per target location? How sensitive is accuracy to F?

---

> ### Author Response · Authors · 2025-11-22
>
> **In this paper, "dynamic" refers to the sensor network changing over time, while the observation distribution remains stationary. The authors should clarify this distinction in the introduction.**
>
> Answer: We adapted our introduction to reflect that in the revision in Line 33-34.
>
> **Spatial cross-attention can be computationally expensive since the method uses all observations to predict the target location. The author has handled the temporal computation cost with trend+seasonality representation, while didn't provide a solution for spatial overhead.**
>
> Answer: We preserved the full spatial structure rather than applying any form of spatial compression to maintain explainability. The model produces an attention matrix between the target and input sensors, indicating the attention weights i.e., the relative importance of each input sensor with respect to each target virtual sensor. In doing so, we intentionally traded additional computational cost for improved interpretability.
>
> **In the experimental section, although the paper claims to provide probabilistic predictions, it only reports CRPS. Additional calibration metric could be added to provide a more comprehensive evaluation.**
>
> Answer: We calculated the Mean Interval Score (MIS) added to the revision of the paper as another metric for the probabilistic predictions in Table - 4.
>
> **Line 420 states "DynaSTI leads on AWN and NACSE," but this appears inconsistent with Table 4. Could the authors clarify which is correct?**
>
> Answer: Table - 4 is correct. It will be NACSE and PEMS-BAY. Thank you for catching our mistake.
>
> **What are the average iteration counts and wall-clock times for the Fourier coefficient fitting at inference per target location? How sensitive is accuracy to F?**
>
> Answer: The iteration count for the Fourier coefficient fitting is reported in the hyperparameters table in Appendix - A.4, Table - 10. You can see it in the Fourier section of the table as the name “iterations”. The wall-clock times have been reported in the Appendix of the revised paper. An  F vs RMSE plot and F vs inference time are also added in the Appendix Figures - 6 and 7, respectively, in the revision.

---

### Official Review · Reviewer_nD44 · 2025-10-31

**Soundness:** 2
**Presentation:** 2
**Contribution:** 1
**Rating:** 2
**Confidence:** 4

**Summary:**

The paper proposes a diffusion model for virtual sensing in inductive sensing. The proposed design appears reasonable and achieves fairly good empirical results on a selection of datasets. I have reviewed a previous version of this paper for a different venue, and I commend the authors for improving the discussion of related work. However, the technical novelty of the paper remains limited, and its contributions are still unclear.

**Strengths:**

- The proposed model provides a reasonable approach for performing missing sensor inference in spatiotemporal data.
- The empirical results are good, but there are missing baselines and insufficient details on how the included baselines were tuned.

**Weaknesses:**

- **Unclear contribution and limited technical novelty.** The contributions of the paper relative to the state of the art remain unclear.
    - The authors summarize the properties of their model in Table 1, but the table shows that a model with analogous characteristics could easily be obtained by combining existing components. In particular, diffusion models for spatiotemporal virtual sensing already exist [1]. The authors claim that their model compares favorably to [1] because the latter uses short input sequences and a fixed graph. However, these limitations can be easily resolved with existing techniques; therefore, the technical novelty of the paper appears limited.
    - In line 72, the paper states: “While USTD meets all criteria in Table 1, the public implementation restricts sequences to 12 or 24 steps, so we were unable to run it on our datasets, which have longer sequences.” Adjusting the input sequence length should not be difficult and should not prevent a direct comparison.
    - The reference for USTD is incorrect; it should refer to [1], not to “Tang et al.”
    - As mentioned above, diffusion models have already been applied to virtual sensing. Likewise, the idea of learning representations in the frequency domain is common in existing methods—for example, [2] combines both diffusion and frequency-domain representations for spatiotemporal forecasting. Therefore, the technical novelty and contributions of this paper appear limited and poorly explained.

- **Empirical evaluation.**
    - How were the baselines tuned for the empirical evaluation? How were the hyperparameters selected?
    - Table 7 reports inference time, but what about training time?


[1] Hu et al., "Towards Unifying Diffusion Models for Probabilistic Spatio-Temporal Graph Learning", SIGSPATIAL 2024

[2] Lin et al., "SpecSTG: A Fast Spectral Diffusion Framework for Probabilistic Spatio-Temporal Traffic Forecasting", arxiv 2024

**Questions:**

Please see comments above.

---

> ### Author Response · Authors · 2025-11-22
>
> **The authors summarize the properties of their model in Table 1, but the table shows that a model with analogous characteristics could easily be obtained by combining existing components. In particular, diffusion models for spatiotemporal virtual sensing already exist [1]. The authors claim that their model compares favorably to [1] because the latter uses short input sequences and a fixed graph. However, these limitations can be easily resolved with existing techniques; therefore, the technical novelty of the paper appears limited.
> In line 72, the paper states: “While USTD meets all criteria in Table 1, the public implementation restricts sequences to 12 or 24 steps, so we were unable to run it on our datasets, which have longer sequences.” Adjusting the input sequence length should not be difficult and should not prevent a direct comparison.**
>
> Answer: We attempted to adapt USTD [1] for our longer time-series data but were unable to make it work. In USTD, the GWaveNetEncoder component compresses the time-series length to 4, and this value is hard-coded. The subsequent modules assume that the encoder output has a length of 4. To achieve this fixed compression, the encoder applies several convolutional layers.
>
> We experimented extensively with the filter size, number of layers, and padding to compress our sequence length of 288 down to 4. However, achieving such a large reduction requires very large filter sizes, even when increasing the number of convolutional layers. When the filter size exceeds 4, the sequence length becomes smaller than the filter size after a few layers, causing the convolution operations to fail. Conversely, keeping the filter small and only increasing the number of layers also failed to produce the required compression.
>
> We reached out to the authors for clarification, but communication eventually stopped without a resolution. For this reason, we concluded that our model can naturally handle longer time-series without additional modifications, whereas USTD cannot be straightforwardly applied in this context.
>
> Our primary contribution lies in unifying all the features in Table 1 within a single model that can seamlessly support any inference scenario, eliminating the need for retraining.
>
> **As mentioned above, diffusion models have already been applied to virtual sensing. Likewise, the idea of learning representations in the frequency domain is common in existing methods—for example, [2] combines both diffusion and frequency-domain representations for spatiotemporal forecasting. Therefore, the technical novelty and contributions of this paper appear limited and poorly explained.**
>
> Answer:  While USTD also adopts a diffusion-based framework, its applicability is limited because its architecture cannot accommodate long time horizons. The public implementation supports only short sequences. In contrast, our model is explicitly designed to handle long sequences without modification.
>
> Similarly, SpecSTG introduces spectral convolution to transform the input graph into the frequency domain, jointly training this spectral representation with the rest of the network. Our approach is fundamentally different: we design a separate Fourier-coefficient fitting module that compresses each input time series by mapping it into the frequency domain independently from the diffusion model. This dedicated module learns trend and seasonality components and then feeds the compressed representation into the diffusion model, enabling efficient long-horizon inference.
>
> To the best of our knowledge, applying this form of explicit Fourier compression as a standalone preprocessing stage for spatio-temporal interpolation, while keeping the generative diffusion model decoupled from the spectral fitting, is the first approach of its kind in this domain.
>
> **How were the baselines tuned for the empirical evaluation? How were the hyperparameters selected?**
>
> Answer: The baselines were tuned by a grid search on the possible choices of the hyperparameters. And the final selected hyperparameters are reported in Appendix A.4.
>
> **Table 7 reports inference time, but what about training time?**
>
> Answer: Training time comparison table (Appendix - A.7, Table - 18) has been added to the revision of the paper.

---

### Meta-Review · Area_Chair_muVz · 2026-01-06

**Summary:**

This paper proposed a diffusion based generative model named DynaSTI to interpolate the patio-temporal data with dynamic sensor sets. Four reviewers provided detailed comments, and 3 of them gave reject (2), one gave marginally above the acceptance threshold (6). The reviewers have a lot of concerns on the limited novelty of the work, the less convincing experiment result and the high computational complexity of the model. Based on their comments, I recommend to reject the paper.

**Reviewer Concerns:**

Addressed:
Reviewer nD44: Model inference time and parameter tuning
Reviewer Wog1: Term clarification
Reviewer fN4f: Lack of sufficient description on data
Reviewer wtC9: substantial computational overhead, more experiment on large datasets

Outstanding:
Reviewer nD44: Unclear contribution and limited technical novelty
Reviewer Wog1: lack of a solution for spatial overhead
Reviewer fN4f: typos

**Reviewer Scores:**

All the reviewers did not gave updated comments based on the rebuttal of the authors. I think it is less likely for the reviewers to change their scores based on the current rebuttal provided.

---

### Decision · Program_Chairs · 2026-01-26

Reject